# Reduction in Absolute Neutrophil Counts in Patient on Clozapine Infected with COVID-19

**DOI:** 10.3390/ijerph182111289

**Published:** 2021-10-27

**Authors:** Fitri Fareez Ramli, Adli Ali, Syed Alhafiz Syed Hashim, Yusof Kamisah, Normala Ibrahim

**Affiliations:** 1Department of Pharmacology, Faculty of Medicine, Universiti Kebangsaan Malaysia, Kuala Lumpur 56000, Malaysia; alhafizsyed@gmail.com (S.A.S.H.); kamisah_y@yahoo.com (Y.K.); 2Department of Psychiatry, University of Oxford, Oxford OX3 7JX, UK; 3Department of Pediatrics, Faculty of Medicine, Universiti Kebangsaan Malaysia, Kuala Lumpur 56000, Malaysia; adli.ali@ppukm.ukm.edu.my; 4Infection and Immunology Health and Advanced Medicine Cluster, Universiti Kebangsaan Malaysia, Kuala Lumpur 56000, Malaysia; 5Department of Psychiatry, Faculty of Medicine and Health Sciences, Universiti Putra Malaysia, Serdang 43400, Malaysia; normala_ib@upm.edu.my

**Keywords:** clozapine, coronavirus, schizophrenia

## Abstract

Despite its severe adverse effects, such as agranulocytosis, clozapine is the primary treatment for treatment-resistant schizophrenia. The established clozapine monitoring system has contributed to reducing agranulocytosis incidence and mortality rates. However, the pandemic coronavirus disease 2019 (COVID-19) has caused changes in the monitoring system. This review aimed to assess the current evidence on the neutrophil changes in the patient on clozapine treatment and infected with COVID-19. Individual cases reported various absolute neutrophil count (ANC) levels, normal, reduced, or elevated. No agranulocytosis case was reported. One case had a borderline moderate-severe ANC level, but the patient was in the 18-week period of clozapine treatment. A cumulative analysis of case the series initially reported inconclusive results. However, a more recent study with a larger sample size reported a significant reduction in the ANC during COVID-19 infection. Nevertheless, this effect is transient as no significant difference was found between the baseline and the post-infection period in ANC levels. In conclusion, COVID-19 is associated with a temporary reduction in ANC levels. The results supported the recommendation to reduce the frequency of clozapine monitoring in the eligible candidates. However, more data are required to confirm the current findings given the limitations, including study design, sample size, and statistical analysis.

## 1. Introduction

Clozapine is the gold standard for treatment-resistant schizophrenia [1]. It was discovered over a half-century ago by researchers at Wander AG, a Swiss pharmaceutical company [2,3]. After a decade, the use of clozapine began to fade away due to its alarming side effect but was re-introduced in the year 1989 for the treatment of refractory psychosis [3]. Numerous comparative studies between clozapine and other antipsychotics have reported superior efficacy of clozapine in treatment-resistant schizophrenia [4,5].

Albeit its robust efficacy, clozapine is underutilized, with less than one-fifth of clozapine-eligible candidates receiving the treatment [2]. The use is limited due to various adverse effects, including agranulocytosis, a severe form of neutropenia characterized by neutrophil counts of less than 0.5 × 10^9^/L [6]. The incidence of agranulocytosis varies across countries, ranging from 0.21 to 0.8% [6]. The mechanisms of clozapine-induced neutropenia and agranulocytosis are not well-understood but may be partly attributable to the selective effect of clozapine on stromal cells and neutrophils. When clozapine is bioactivated to cytotoxic metabolites, cell death is promoted secondary to oxidative stress and haptenation [7,8,9].

The establishment of a clozapine monitoring system allows for early detection of severe agranulocytosis [10], enabling timely intervention to be administered and thus reducing agranulocytosis incidence and mortality rates [11]. Although the established monitoring system has been successfully conducted for many years, the global pandemic of coronavirus disease 2019 (COVID-19) has forced significant adjustments to ensure clozapine treatment continuation while reducing COVID-19 exposure to patients. A consensus on clozapine was published to guide clinicians with focus on the frequency of ANC monitoring, clinical assessment, and dose changes [12]. In terms of ANC monitoring, the ANC frequency can be reduced to three monthly if the patient has been on clozapine treatment for more than a year with no history of ANC of less than 2000/µL (or less than 1500/µL if the patient has history of benign ethnic neutropenia), and no practical or safe access to ANC testing [12].

At the time this consensus was published, limited data were available regarding the effects of COVID-19 in patients receiving clozapine. In the general population, COVID-19 has been reported to exert a significant effect on hematological parameters. Asghar, et al. [13] reported a significantly lower white blood cell count in COVID-19 patients than healthy people. Evidence from various studies demonstrated that lymphopenia is a more common finding in COVID-19 patients than other hematological parameters [14]. Moreover, neutrophilia is an essential predictor for poor prognosis in severe COVID-19-infected patients [15,16]. The release of pro-inflammatory mediators, such as interleukin (IL-6, IL-10) following neutrophil infiltration, contributes to ‘cytokine storms’ with the subsequent hyperinflammation state [15,16]. The infiltration of inflammatory cells in the lung causes massive injury that might attribute to acute respiratory distress syndrome [15]. On the contrary, growing evidence has reported a reduction in ANC in COVID-19 patients receiving clozapine [17,18,19]. The magnitude of ANC reduction is unclear during COVID-19 in patients receiving clozapine. Moreover, the association of COVID-19 infection with neutropenia and agranulocytosis incidence in this population is unknown.

This review aimed to assess the current evidence on the neutrophil changes in patients on clozapine treatment who were infected with COVID-19. The findings are essential to inform the clinicians regarding the risk or incidence of neutropenia and agranulocytosis in clozapine-treated patients as the new recommendation has changed the monitoring procedure in order to reduce exposure to COVID-19. We also revisited the mechanism of clozapine-induced neutropenia in the initial section.

## 2. The Mechanisms of Clozapine-Induced Neutropenia

Clozapine has been reported to exert initial bone marrow stimulation with subsequent risk of neutropenia, particularly between 6–18 weeks of treatment [20,21]. A neutrophil kinetic study demonstrated a gradual, but significant increase in blood neutrophil levels [21,22] with a concurrent increase in the granulocyte colony-stimulating factor (G-CSF) at three and six hours following clozapine administration [21]. These findings indicate kinetics of neutrophils are G-CSF-dependent. Interestingly, the clozapine effect is only restricted to neutrophils but not lymphocytes [21,22]. These findings were further supported by clinical findings that showed a transient elevation of neutrophil levels in a cohort of 100 patients in the first few weeks of clozapine treatment [17]. This transient elevation in the neutrophil count subsequently normalizes, however in a smaller cohort of patients, clozapine eventually induced neutropenia or agranulocytosis. The exact pathology underlying to this phenomenon is not fully elucidated, nonetheless several mechanisms have been proposed and a significant genetic predisposition has been observed.

One of the mechanisms that was initially proposed is related to the direct toxicity of clozapine metabolites to the bone marrow stromal cells, in particular the immature neutrophil subpopulation. It is worth noting that, clozapine in its natural form itself is cytotoxic, but only at supratherapeutic levels [9]. The metabolism of clozapine forms at least three different metabolites with distinct cytotoxicity levels (Figure 1). Clozapine *N*-oxide is a non-toxic metabolite, while *N*-desmethylclozapine is a less toxic metabolite than clozapine. Both are the products of a cytochrome P450 enzyme (CYP) metabolism, particularly *CYP3A4* and *CYP1A2* [23].

Nitrenium ion is a reactive and toxic metabolite of clozapine produced by the interaction of clozapine with hypochlorous acid (HOCI), a primary oxidant found in the activated neutrophils (Figure 2) [24]. The formation of nitrenium ion (Figure 3) can also be catalyzed by a combination of horseradish peroxidase and hydrogen peroxide (H_2_O_2_), as well as a combination of myeloperoxidase, H_2_O_2_, and chloride ion [8,9,24,25]. Interestingly, nitrenium ion formation is specific to clozapine. Utilizing the same metabolizing system (horseradish peroxidase and hydrogen peroxide) for other antipsychotics such as olanzapine and risperidone did not produce toxic metabolites [7]. Nitrenium ion has the ability to form a covalent bond with neutrophils, forming neutrophil haptenation which promotes apoptosis via tyrosine kinase activation [9,25].

Antioxidants play a pivotal role in the detoxification of toxic metabolites. Glutathione (GSH), an antioxidant, detoxifies nitrenium ions via conjugation [25]. Other exogenous antioxidants such as ascorbic acid, catalase, and N-acetylcysteine can also reduce clozapine-mediated cytotoxicity in stromal cells and neutrophils, further supported by the mechanism of neutropenia of the drug via oxidative stress [7,8].

Although the neutropenic effect of clozapine is consistently observed in in vitro studies, the prevalence of neutropenia and agranulocytosis in patients receiving clozapine is considerably low globally. The advancement of pharmacogenomics technologies has allowed the determination of genetic factors at molecular levels. Studies from few populations have reported significant associations of genetic polymorphisms in human leukocyte antigen (HLA), ABCB1 drug transporter, and glutathione S-transferase with clozapine-induced neutropenia [26,27]. Moreover, genetic polymorphisms in HLA, ABCB1, and NRH-quinone oxidoreductase 2 have been found to demonstrate significant associations with clozapine-induced agranulocytosis [26,27,28]. Additionally, another recent study also identified the *STARD9* and *UBAP2* gene variants in a cohort of patients with clozapine-induced neutropenia. *STARD9* is a mitotic kinesin, essential for pericentriolar matrix cohesion in mitosis. Depletion of *STARD9* promotes mitotic arrest secondary to fragmentation and dissociation of pericentriolar material, resulting in apoptosis in various cancer cells [29]. The *UBAP2* gene regulates ubiquitination, an essential process involved in cell proliferation and survival [30]. The combined effect of the gene variants and the direct toxic effect by the clozapine’s metabolites promotes neutrophil toxicity [31]. However, given the statistical and methodological limitations, further studies are required to justify the role of genetic variants in clozapine-induced neutropenia and agranulocytosis.

## 3. COVID-19 Infection and Neutrophil Count in Patients on Clozapine Treatment

### 3.1. Evidence from Case Series

Initial data from the case series demonstrated inconclusive findings from two centers. Gee and Taylor [18] reported a case series of 13 patients on clozapine that were infected with COVID-19 with various underlying conditions, including diabetes mellitus, hypertension, and anemia (Table 1). A cumulative analysis of patients revealed a declining trend in ANC during the infection but an increasing trend in the post-infection period. However, negligible differences were reported between baseline and during infection, and infection and post-infection. Interestingly, there was no significant difference in ANC between baseline and post-infection. Moreover, out of 13 patients, only one patient had ANC of less than 1.5 × 10^9^/L, and no agranulocytosis case was reported. Unfortunately, two patients died from COVID-19, while another patient died from myocardial infarction. These patients were reported to have multiple comorbidities.

On the other hand, Bonaccorso, et al. [17] found a significant reduction and improvement during and post-COVID-19 infection, respectively, among ten patients infected with COVID-19. In this cohort, no mortality was reported, and only one patient was admitted to the intensive care unit due to complications. Another case series [33] is described in the following subtopic as the authors did not analyze the cases cumulatively but described them individually.

The different findings between the case series might be attributable to several factors. The definitions of the pre-, during, and post-COVID-19 periods are different between the two case series. Gee and Taylor [18] defined 14 days before the positive swab test, 0–7 days, and 8–14 days as the pre-, during, and post-COVID-19 infection period, respectively. On the contrary, Bonaccorso, et al. [17] defined the period based on the average ANC in the previous three years, around three days (±2.83 days) after the onset of the symptoms, and approximately one month after the COVID-19 infection, as pre-, during, and post-COVID-19, respectively. Another reason is probably the difference in the statistical analysis used between studies, with one study utilized a paired *t*-test [18], while another used the linear regression method [17]. However, none of the studies used a multiple regression model to control for specific confounding factors. Other factors may be related to different proportions and types of comorbidities between the case series.

A more recent study that combined datasets from both studies [17,18] with additional samples reported a significant reduction in the ANC level during COVID-19 infection [19]. Furthermore, the post-COVID-19 ANC values were not significantly different from the baseline. In this more extensive study, the statistical analysis method used was similar to those described by Gee and Taylor [18] that initially only found specific trends but no significant results.

### 3.2. Evidence from Case Report

Evidence from numerous cases reported various ANC levels (normal, reduced, or elevated) in patients on clozapine infected with COVID-19. Butler, et al. [33] reported a patient with a normal ANC value during the infection. This patient had been on clozapine treatment for more than a year and was maintained on 600 mg daily. Unfortunately, the patient did not survive as he was complicated with COVID-19 pneumonia. Boland and Dratcu [36] reported a case of normal ANC in a patient who has been on clozapine treatment for 18 months. Initially, the patient presented with abnormal vitals (hyperpyrexia, tachycardia, and hypertension) with a deranged renal profile. Eventually, the patient condition improved and they were discharged. Moreover, Dotson, et al. [35] reported a patient maintained on clozapine for many years with a normal ANC level during COVID-19. This patient had stable ANC levels during her stay and was eventually discharged.

On the other hand, Dotson, et al. [35] reported a decline in ANC level below the normal range with an elevated level of clozapine. In this case, the authors hypothesized that tocilizumab, a medication for COVID-19, might partly explain ANC reduction. Tocilizumab works through the inhibition of interleukin-6, an essential cytokine in neutrophil kinetics [37]. The mechanism of tocilizumab-induced neutropenia is attributable to the inhibition of neutrophil recruitment into peripheral blood and possibly apoptosis induction [37].

Similarly, Cranshaw and Harikumar [34] reported a mild reduction in ANC levels. However, no COVID-19 treatment was started in this case, but the patient was noted to have a significant lymphopenia and signs of clozapine toxicity, leading to a temporary cessation of clozapine. Serum clozapine and norclozapine levels taken reported a notable increase from the baseline values. The possible mechanism for elevated clozapine levels in infection or inflammation is the inhibition of *CYP1A2*, one of the primary clozapine metabolizing enzymes [38]. Cytokines released during infection can downregulate CYP activity, which in turn reduces the clozapine metabolism [39]. The patient was reported to have an uncomplicated COVID-19 recovery.

Moreover, Gee and Taylor [19] reported four cases of reduced ANC value. Two had episodes of mild neutropenia, one had an episode of moderated neutropenia, while another had a borderline episode of moderate-severe neutropenia (ANC value of 0.5 × 10^9^/L). Clozapine treatment was temporarily withheld in all but one, in which clozapine treatment was discontinued permanently. In this case, the patient just started on clozapine (67 days). The follow-up post-COVID-19 infection in all four cases reported that all ANC were normalized.

In contrast, Butler, et al. [33] demonstrated another case of an elevated ANC level. However, no other ANC level was reported, probably because the patient was complicated with a urinary tract infection and still under-recovery. The reason for the elevated ANC might be partly explained by cytokines released during the infection. This fact is supported by the finding from a study in the general population that reported a significant increment of cytokines, such as interleukin-6 and tumor necrosis factor-α in COVID-19 patients compared with controls [40].

### 3.3. Perspectives

The results from the current studies demonstrated ANC might be maintained within normal values, elevated, or reduced during COVID-19 infection. However, the cumulative findings reported a transient but significant reduction during COVID-19 infection. No agranulocytosis case was reported but one patient was borderline. This case is unique as the patient was in the period of 18 weeks of clozapine treatment, when the risk of agranulocytosis is the highest [11]. In the majority of the reports, the ANC value rebounded to baseline post-COVID-19 infection. The data supported the recommendation to reduce the frequency of ANC monitoring for the patients who have been stabilized on clozapine for more than a year without history of low ANC levels (<2000/µL or <1500/µL in patient with BEN) and no practical access to test the ANC levels. This evidence can potentially change the clozapine monitoring system as a whole, even when the pandemic is over in the future.

However, the nature of the current evidence (case report and case series) hinders the assessment of a causal relationship between neutropenia and COVID-19 infection. Moreover, a small sample size prevents more a comprehensive assessment of the association of neutropenia and COVID-19 infection. Moreover, most of the studies are reported from the United Kingdom. The authors only analyzed the data using a univariate test due to a limited sample size. As a result, various confounding factors could not be controlled. Distinct characteristics between individuals in terms of demographic and clinical backgrounds, including age, race, gender, comorbidities, and medications, have the potential to confound the correlation between ANC and COVID-19 infection. The most extensive study (n = 53) included in this review reported a higher proportion of patients aged ≥ 40 years, non-Caucasian group, male predominance, and a wide range of clozapine treatment duration (12 days–28 years) [19]. The risk of clozapine-induced neutropenia is higher in the younger population, Black population, lower WBC baseline, and within 6–18 weeks of clozapine treatment [20]. Other than that, the combination of clozapine and other antipsychotic medication may alter the risk of neutropenia. A recent meta-analysis study reported no significant difference in the risk of neutropenia between clozapine and other antipsychotics, including risperidone and olanzapine [41]. The finding may indicate that other antipsychotics have the potential to cause neutropenia. However, no data are available on the effect or association of other antipsychotics on ANC levels in COVID-19 patients. Future studies should be conducted to investigate the association or temporal relationship between other antipsychotics and ANC in this population. Comparing clozapine and other antipsychotics on the impact of ANC can provide more insights on the risk of neutrophil levels reduction in COVID-19. Moreover, the direction of the future study should focus on the cohort or cross-sectional studies with a larger sample size, controlling the confounding factors, and be replicated in other countries. Despite numerous limitations, current evidence indicates that clozapine use during COVID-19 infection did not result in significant reduction in ANC levels.

## 4. Conclusions

To summarize, COVID-19 is associated with a temporary reduction in ANC levels. The results supported the recommendation to reduce the frequency of clozapine monitoring in eligible candidates. However, more data are required to confirm current findings given the limitations including study design, sample size, and statistical analysis.

## Figures and Tables

**Figure 1 ijerph-18-11289-f001:**
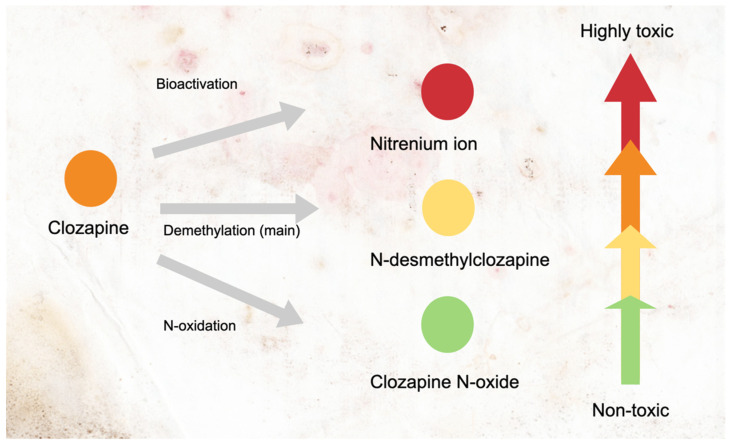
Metabolism of clozapine into metabolites and its cytotoxicity levels. Clozapine is primarily metabolized by cytochrome P450 (CYP) into desmethylclozapine. Another product of CYP metabolism is clozapine *N*-oxide, a non-toxic and inactive metabolite. Some clozapine is converted into nitrenium ions by hypochlorous acid (HOCI), the oxidant found in the activated neutrophils. The arrows with colors indicate the cytotoxicity levels of each compound in increasing order.

**Figure 2 ijerph-18-11289-f002:**
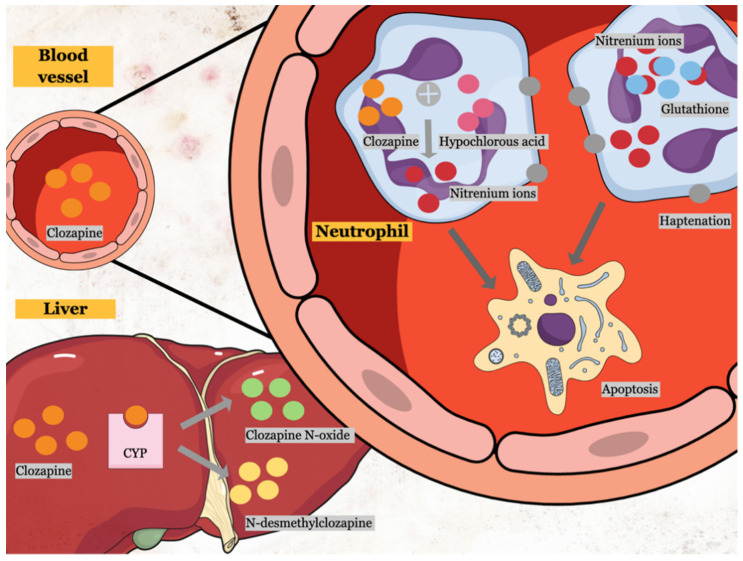
Mechanism of clozapine-induced neutrophil toxicity. Clozapine is metabolized by cytochrome P450 (CYP) into clozapine *N*-oxide and N-desmethylclozapine in the liver. Moreover, clozapine can react with hypochlorous acid in the neutrophils to form nitrenium ions, which later can form haptenation on the membrane surface, promoting apoptosis. Nitrenium ions can be detoxified by antioxidants such as glutathione (GSH) to form non-toxic GSH conjugates. Excessive use of GSH can lead to depletion, increasing cellular oxidative stress, resulting in cell death.

**Figure 3 ijerph-18-11289-f003:**
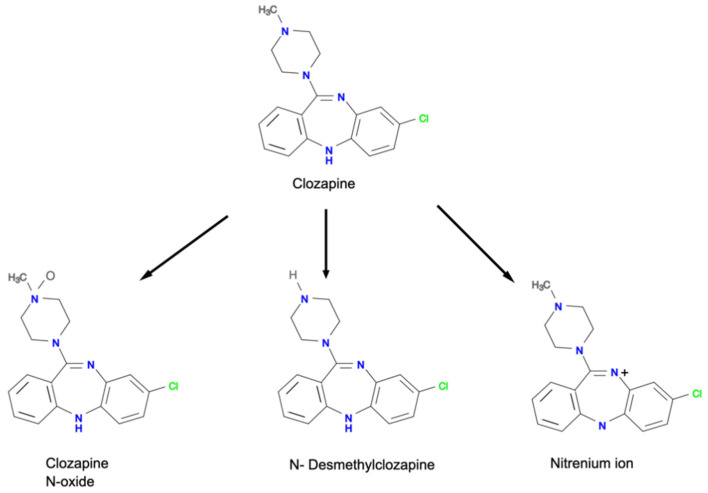
Chemical structures of clozapine and nitrenium ion (adapted from Pirmohamed and Park [32]).

**Table 1 ijerph-18-11289-t001:** Characteristics of case series and case reports included in the article.

References	Population/Patient Characteristics	Duration and Dose of Clozapine	Location	ANC			Remarks
				Baseline	During Infection	Post Infection	
Case series							
Gee [18]	Thirteen patients with a mean of 48 years of age. The majority were women (69%), non-Caucasian (69%) and had comorbidities (85%).	Duration: Less than 18 weeks–more than 1 year (range) Dose: NA	South London and the Maudsley NHS Foundation Trust.	4.83 × 10^9^/L	4.24 × 10^9^/L	5.70 × 10^9^/L	
Bonaccorso [17]	Ten patients with the median of 45.4 years of age. The majority were male (70%), of African descent (50%), and had some comorbidities.	Duration: 726.1 days (mean) Dose: 200–600 mg/day (range)	Highgate Mental Health Centre, Camden and Islington NHS Foundation Trust.	5.20 × 10^9^/L	4.13 × 10^9^/L ↓ *	5.28 × 10^9^/L ↑ **	The clozapine dose from one patient was reduced while another was stopped.
Gee [19]	Fifty-three patients with the age range between less than 20 years of age and more than 80 years of age. The majority were male (64%), non-Caucasian (64%), and had schizophrenia diagnoses (64%).	Duration: 4.6 years (mean) 12 days–28 years (range) Dose: 342 mg/day (mean) 50–800 mg/day (range)	Five centers under the London Mental Health Trusts.	4.72 × 10^9^/L	3.83 × 10^9^/L ↓*	4.73 × 10^9^/L	This datasets comprises Gee and Taylor [18] and Bonaccorso, et al. [17] with additional samples. A total of 36% of patients had their clozapine dose changed/stopped due to various reasons.
Case report							
Butler [33]	A 62-year-old man (African descent) with schizoaffective disorder presented with delirium, fever, respiratory symptoms, and signs. However, the patient had a worsening respiratory condition and died of COVID-19 pneumonia.	Duration: Since late 2018 Dose: 600 mg daily	A university teaching hospital in London, 16 March–1 May 2020.	NA	1.64 × 10^9^/L (N)	NA (patient died)	Clozapine was withheld for less than 24 h then restarted at 400 mg ON.
Butler [33]	A 57-year-old woman (Caucasian) with treatment-resistant schizophrenia presented with hypoxia and hemodynamic instability. The patient was supported with a mechanical ventilator. Subsequently, the patient was in slow recovery at the time the article was written.	Duration: NA Dose: 350 mg daily	A university teaching hospital in London, 16 March–1 May 2020.	NA	10.3 × 10^9^/L ↑	NA	Clozapine was stopped but retitrated on day 19.
Cranshaw [34]	A 38-year-old man with organic psychosis presented with respiratory signs and symptoms and clozapine toxicity signs (hypersalivation and myoclonus). The patient had elevated clozapine (0.73 mg/l) and norclozapine (0.31 mg/l) levels.	Duration: NA Dose: 325 mg daily	NA (UK)	NA	1.26 × 10^9^/L ↓	NA	Clozapine was temporarily stopped due to significant lymphopenia. Uncomplicated recovery.
Dotson [35]	A 76-year-old man with schizoaffective disorder (bipolar-type) with recurrent catatonia on clozapine 300 mg ON and monthly ECT. The patient presented with catatonia. High clozapine levels on admission (1360 ng/mL). Concurrent use with tocilizumab.	Several years	NA	NA	1.10 × 10^9^/L ↓	4.00 × 10^9^/L	
Dotson [35]	A 63-year-old woman with schizoaffective disorder (bipolar-type) on clozapine 50 mg OM/350mg ON, olanzapine 20 mg ON, citalopram 20 mg OD. The patient presented with confusion, nausea, ileus, severe hyponatremia with high clozapine levels on admission (1060 ng/mL).	NA	NA	NA	14.97 × 10^9^/L ↑	NA	Clozapine was withheld for a week and was retitrated as her bowel function normalized.
Dotson [35]	A 53-year-old woman with schizophrenia on clozapine 250 mg ON. The patient presented with delirium, fever, and vomiting. She had high clozapine levels on admission (2154 ng/mL). Her ANC level was stable during her stay. The patient was eventually discharged.	Several years	NA	NA	2.20 × 10^9^/L (N)	NA	Clozapine dose was reduced.
Boland [36]	A 21-year-old man with treatment-resistant schizoaffective disorder on clozapine 850 mg ON and lithium 1.2 g ON for 18 months. He presented with fever, coryzal symptoms, high blood pressure, tachycardia, deranged renal profile. The patient eventually became well and was discharged.	18 months		NA	6.94 × 10^9^/L (N)	NA	Clozapine was withheld for three days and then retitrated over five days to 600 mg daily.
Gee [19]	A non-Caucasian woman in a 21–30 age group was diagnosed with COVID-19 infection (no further information provided).	Duration: 231 days Dose: NA	One of five centers under the London Mental Health Trusts.	>2.50 × 10^9^/L	1.2–1.4 × 10^9^/L (Day 8–9) ↓	NA (reported as resolved)	Clozapine was withheld for 24 h.
Gee [19]	An African descent man in the 51–60 age group with a history of moderate neutropenia (0.7 × 10^9^/L ANC) when he was under critical care due to a motor vehicle accident.	Duration: 520 days Dose: NA		NA	0.9 × 10^9^/L (Day 5), 1.2–2.1 × 10^9^/L (Day 6–8) ↓	>2.0 × 10^9^/L	Clozapine was withheld (no specific duration reported) but then restarted.
Gee [19]	A Caucasian man in a 31–40 age group.	Duration: 339 days Dose: NA		NA	>2.5 × 10^9^/L (until day 5) 1.2 × 10^9^/L (Day 6) ↓	2.7 × 10^9^/L (Day 8)	Clozapine was withheld but restarted on day 10.
Gee [19]	A non-Caucasian man in a 21–30 age group.	Duration: 67 days Dose: NA		≥2.9 × 10^9^/L	0.5–0.9 × 10^9^/L ↓	NA (but resolved on day 38)	Clozapine treatment stopped.

* Significantly different between during infection vs. baseline; ** significantly different between post-infection vs. during infection; ↓ decrease; ↑ increase; ANC: absolute neutrophil count; COVID-19: coronavirus disease 2019; ECT: electroconvulsive therapy; BD: twice daily; NA: not available; NS: not significant; OM: on morning; OD: once daily; ON: on night.

## Data Availability

Not applicable.

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
