# Peer review of "Reduction in Absolute Neutrophil Counts in Patient on Clozapine Infected with COVID-19"

_ijerph, 2021, doi:10.3390/ijerph182111289_

Round 1

Reviewer 1 Report

Very interesting and topical work.
Very well exposed. Good level of English.
Fiugres and Tables are ok. Congratulations!

More than 50% of bibliographic citations are recent
s.

The introduction lacks a brief review on the prevalence of neutropenia-agranulocytosis related to taking clozapine. P.e granulocytosis occurs in up to 0.8% of patients and presents a significant medical challenge, despite decreasing mortality rates (Mijovic a, 2020 - cite 6)
Perhaps the limitations of the study should not be included in the conclusions section but should be included in section 3

Author Response

No

Comments

Replies

Reviewer 1

1.

Very interesting and topical work.
Very well exposed. Good level of English.
Figures and Tables are ok. Congratulations!

More than 50% of bibliographic citations are recents.

Thank you very much for your valuable feedback.

2.

The introduction lacks a brief review on the prevalence of neutropenia-agranulocytosis related to taking clozapine. P.e granulocytosis occurs in up to 0.8% of patients and presents a significant medical challenge, despite decreasing mortality rates (Mijovic a, 2020 - cite 6)

Thank you very much for your valuable feedback. We have included the incidence of agranulocytosis in the introduction section.

Page no. 1, line no. 43-44.

“The incidence of agranulocytosis varies across countries, ranging from 0.21% - 0.8% [6].”

3.

Perhaps the limitations of the study should not be included in the conclusions section but should be included in section 3

Thank you very much for your valuable feedback.

We have rearranged our perspective that was initially placed under “Conclusion and Perspectives” to Section 3 as suggested under “3.3. Perspectives”. Page no. 10-11, line no. 292-330.

Reviewer 2 Report

This well-written review collates and describes the current evidence with respect to the effect of COVID-19 on clozapine-induce neutropenia and agranulocytosis. A sound comparison of the two largest studies, which have divergent results, is included. Given the evidence that patients are genetically predisposed to severe clozapine-induce neutropenia and granulocytosis, the conclusion supporting the recommendation to reduce the frequency of ANC monitoring during the pandemic, in stable patients on clozapine for more than 1 year with no practical or safe testing center, is supported by the evidence presented.

Author Response

No

Comments

Replies

1.

This well-written review collates and describes the current evidence with respect to the effect of COVID-19 on clozapine-induce neutropenia and agranulocytosis. A sound comparison of the two largest studies, which have divergent results, is included. Given the evidence that patients are genetically predisposed to severe clozapine-induce neutropenia and granulocytosis, the conclusion supporting the recommendation to reduce the frequency of ANC monitoring during the pandemic, in stable patients on clozapine for more than 1 year with no practical or safe testing center, is supported by the evidence presented.

Thank you very much for your valuable feedback. We appreciate your effort and time to review this article.

Reviewer 3 Report

The paper of Ramli et al. reports on a short review of the reduction of Absolute Neutrophil counts (ANC) in clozapine-treated patients infected with COVID-19. I recommend the publication of this paper in the International Journal of Environmental Research and Public Health (IJERPH); with major revision specially in the report of the participants replies according to the comments that follow:

    The title could be revised to better illustrate the main purpose of the review, as well as the abstract.
    The review could bring more insight about the consequences of other schizophrenia-treatments and derivatives such as olanzapine, risperidone was very briefly described.
    In the Introduction, the background could be further presented and the impact of interleukine effect on ANC introduced.
    62: the authors should check the reference and authors.
    L130: the authors could better explain the significance of the STARD9 and UBAP2 identified genes.
    The units should be revised in the table and expressed in L.
    The analysis shown in the table seems to be biased: the patients have indeed different clinical cases, and history of schizophrenia treatment (the causes related to the change/end of clozapine vary as well).
    The review requires deeper analysis to better understand the impact and correlation with the COVID - infection.

The authors could have shown the chemical structure of clozapine to better explain the chemical reactions leading to the formation of toxic metabolites.

Author Response

No

Comments

Replies

1.

The paper of Ramli et al. reports on a short review of the reduction of Absolute Neutrophil counts (ANC) in clozapine-treated patients infected with COVID-19. I recommend the publication of this paper in the International Journal of Environmental Research and Public Health (IJERPH); with major revision specially in the report of the participants replies according to the comments that follow

Thank you very much for your valuable feedback. We have included the following points to improve our articles according to your suggestions.

2.

The title could be revised to better illustrate the main purpose of the review, as well as the abstract.

Thank you very much for your valuable feedback.

We have changed our title according to your suggestion:

“Reduction in Absolute Neutrophil Counts in Patient on Clozapine Infected with COVID-19”.

3.

The review could bring more insight about the consequences of other schizophrenia-treatments and derivatives such as olanzapine, risperidone was very briefly described.

Thank you very much for the valuable feedback. Although this is interesting subtopic to explore, the evidence of other antipsychotics association or its impact on absolute neutrophil count is scarced. However, given the importance to address this issue, we have included some descriptiom and suggestion on this matter.

Page no. 11, line no. 317-326:

“Other than that, the combination of clozapine and other antipsychotic medication may alter the risk of neutropenia. A recent meta-analysis study reported no significant difference in the risk of neutropenia between clozapine and other antipsychotics, including risperidone and olanzapine [41]. The finding may indicate that other antipsychotics have the potential to cause neutropenia. However, no data is available on the effect or association of other antipsychotics on ANC levels in COVID-19 patients. Future studies should be conducted to investigate the association or temporal relationship between other antipsychotics and ANC in this population. Also, comparing clozapine and other antipsychotics on the impact of ANC can provide more insights on the risk of neutrophil levels reduction in COVID-19.”

4.

In the Introduction, the background could be further presented and the impact of interleukine effect on ANC introduced.

Thank you very much for the valuable feedback. We have added the following statement describing the impact of interleukin on ANC:

Page no. 2, line no. 76-81

“Moreover, neutrophilia is an essential predictor for poor prognosis in severe COVID-19-infected patients [15,16]. The release of pro-inflammatory mediators, such as interleukin (IL-6, IL-10) following neutrophil infiltration, contributes to ‘cytokine storms’ with the subsequent hyperinflammation state [15,16]. The infiltration of inflammatory cells in the lung causes massive injury that might attribute to acute respiratory distress syndrome [15].”

5.

62: the authors should check the reference and authors.

Thank you very much for your valuable feedback. We have checked the reference and confirmed our statement.

6.

L130: the authors could better explain the significance of the STARD9 and UBAP2 identified genes.

Thank you very much for your valuable feedback. We have added additional description of STARD9 and UBAP2.

Page no. 4, line no. 148-153

STARD9 is a mitotic kinesin, essential for pericentriolar matrix cohesion in mitosis. Depletion of STARD9 promotes mitotic arrest secondary to fragmentation and dissociation of pericentriolar material, resulting in apoptosis in various cancer cells [29]. UBAP2 gene regulates ubiquitination, an essential process involved in cell proliferation and survival [30]. The combined effect of the gene variants and the direct toxic effect by the clozapine’s metabolites promotes neutrophil toxicity [31].”

7.

The units should be revised in the table and expressed in L.

Thank you very much for your valuable feedback.

We have changed the unit from ‘l’ to ‘L’. Kindly refer to Table 1, page no. 6-9.

8.

The analysis shown in the table seems to be biased: the patients have indeed different clinical cases, and history of schizophrenia treatment (the causes related to the change/end of clozapine vary as well). The review requires deeper analysis to better understand the impact and correlation with the COVID - infection.

Thank you very much for your valuable feedback. We have elaborated further on this issue:

Page no. 10-11, line no. 310-326.

“Distinct characteristics between individuals in terms of demographic and clinical backgrounds, including age, race, gender, comorbidities, and medications, have the potential to confound the correlation between ANC and COVID-19 infection. The most extensive study (n=53) included in this review reported a higher proportion of patients aged > 40 years, non-Caucasian group, male predominance, and a wide range of clozapine treatment duration (12 days – 28 years) [19]. The risk of clozapine-induced neutropenia is higher in the younger population, Black population, lower WBC baseline, and within 6-18 weeks of clozapine treatment [20]. Other than that, the combination of clozapine and other antipsychotic medication may alter the risk of neutropenia. A recent meta-analysis study reported no significant difference in the risk of neutropenia between clozapine and other antipsychotics, including risperidone and olanzapine [41]. The finding may indicate that other antipsychotics have the potential to cause neutropenia. However, no data is available on the effect or association of other antipsychotics on ANC levels in COVID-19 patients. Future studies should be conducted to investigate the association or temporal relationship between other antipsychotics and ANC in this population. Also, comparing clozapine and other antipsychotics on the impact of ANC can provide more insights on the risk of neutrophil levels reduction in COVID-19.”

9.

The authors could have shown the chemical structure of clozapine to better explain the chemical reactions leading to the formation of toxic metabolites.

Thank you very much for your valuable feedback. We haved added figure 3 that contains chemical structures of clozapine and its metabolites.

Page no. 5.

Round 2

Reviewer 3 Report

The authors have properly responded to the suggested comments and I recommend the paper to be published in the present form.